# FedUP: Querying Large-Scale Federations of SPARQL Endpoints

## ABSTRACT

Processing SPARQL queries over large federations of SPARQL endpoints is crucial for keeping the Semantic Web decentralized. Despite the existence of hundreds of SPARQL endpoints, current federation engines only scale to dozens. One major issue comes from the current definition of the source selection problem, i.e., finding the minimal set of SPARQL endpoints to contact per triple pattern. Even if such a source selection is minimal, only a few combinations of sources may return results. Consequently, most of the query processing time is wasted evaluating combinations that return no results. In this paper, we introduce the concept of Result-Aware query plans. This concept ensures that every subquery of the query plan effectively contributes to the result of the query. To compute a Result-Aware query plan, we propose FedUP, a new federation engine able to produce Result-Aware query plans by tracking the provenance of query results. However, getting query results requires computing source selection, and computing source selection requires query results. To break this vicious cycle, FedUP computes results and provenances on tiny quotient summaries of federations at the cost of source selection accuracy. Experimental results on federated benchmarks demonstrate that FedUP outperforms state-of-the-art federation engines by orders of magnitude in the context of large-scale federations.

**ACM Reference Format:**
Anonymous Author(s). 2023. FedUP: Querying Large-Scale Federations of SPARQL Endpoints. In *Proceedings of The Web Conference (WWW'24)*. ACM, New York, NY, USA, 10 pages. https://doi.org/10.1145/nnnnnnn.nnnnnnn

## 1 INTRODUCTION

***Context and motivation.*** Processing SPARQL queries over large federations of SPARQL endpoints is crucial for keeping the Semantic Web decentralized. Despite the existence of hundreds of SPARQL endpoints [13, 24], current federation engines [6, 19, 20, 22] only scale to dozens [9]. This is a severe issue for developing an effective, usable, and decentralized Semantic Web based on federation engines and federations of SPARQL endpoints.

***Related work and problem:*** Federated query processing has 3 conceptual steps [2]:

(1) source selection and query decomposition
(2) query optimization
(3) query execution

```
SELECT DISTINCT ?product ?localProdLabel WHERE {
  ?lProd  rdfs : label  ?lProdLabel  .                        #tp1 @ rs6,  rs0
  ?lProd  bsbm:productFeature ?lProdFeature  .                #tp2 @ rs6,  rs0
  ?lProd  bsbm:productPropertyNumeric1 ?simProperty1 .        #tp3 @ rs6,  rs0
  ?lProd  bsbm:productPropertyNumeric2 ?simProperty2 .        #tp4 @ rs6,  rs0
  ?lProd  owl:sameAs ?product  .                              #tp5 @ rs6,  rs0
  ?lProdFeature  owl:sameAs ?prodFeature  .                   #tp6 @ rs6,  rs0
  ?lProdXYZ bsbm:productFeature ?lProdFeatureXYZ  .           #tp7 @ v3,   v3
  ?lProdXYZ bsbm:productPropertyNumeric1 ?origProperty1 .     #tp8 @ v3,   v3
  ?lProdXYZ bsbm:productPropertyNumeric2 ?origProperty2 .     #tp9 @ v3,   v3
  ?lProdXYZ owl:sameAs bsbm:Product136030 .                   #tp10@ v3,   v3
  ?lProdFeatureXYZ owl:sameAs ?prodFeature  .                 #tp11@ v3,   v3
  FILTER  (...)} ORDER BY ?lProdLabel LIMIT 5
```

**Figure 1: Cross Domain Query $q05$ of FedShop [9] along with its optimal source selection over a federation of 20 shops.**

**Table 1: $q05$ execution times using FedShop's reference, a state-of-the-art federation engine, and our proposal FedUP.**

|  | 20 shops | 200 shops |
|---|---|---|
| RSA | 50ms | 1.5s |
| CostFed | 2.45s | > 1h |
| **Our proposal (FedUP)** | **244ms** | **12.4s** |

One major issue comes from the current definition of the *source selection problem*, i.e., finding the minimal set of SPARQL endpoints to contact per triple pattern [19]. Even if such a source selection is minimal, only a few combinations of sources may return results. Consequently, most of the query processing time is wasted evaluating combinations that return no results. To illustrate, Figure 1 presents the query $q05$ of the FedShop benchmark [9], along with the set of sources to contact per triple pattern. Triple patterns that share the same single data source are merged into exclusive groups [22], e.g., $tp7 - tp11$ are grouped together to be executed on $v3$.

Then, the objective of the optimizer is to generate an execution plan that minimizes the number of intermediate results and the communication costs. Thanks to heuristics and/or statistics, it can decide a particular join order and physical operators. In order to avoid huge data transfer of general predicates such as the sameAs predicate in $tp4$ and $tp5$, the query optimizer may decide a BoundJoin [22].

Finally, during query execution, a physical query plan for $q05$ based on relevant source per triple pattern and BoundJoin operator checks every 64 combinations of sources even when 2 combinations only effectively return results:

(1) $[rs6, rs6, rs6, rs6, rs6, rs6, v3, v3, v3, v3, v3]$
(2) $[rs0, rs0, rs0, rs0, rs0, rs0, v3, v3, v3, v3, v3]$

As the number of useless combinations increases with federation size, state-of-the-art federation engines suffer from serious performance issues as reported in Table 1. While FedShop empirically

**Figure 2: Federation $F_1$ with 4 endpoints storing information about Scorpions and Kraftwerk, 2 bands from Germany.**

demonstrates that an engine could evaluate $q05$ in less than $2s$ (RSA), CostFed, the best federation engine on FedShop, needs $3s$ to finish evaluating $q05$ with 20 sources and more than one hour with 200 sources [1]. Our proposal FedUP can process $q05$ in $12.4s$ on the federation of 200 endpoints.

**Approach and contributions:** In this paper, we introduce the concept of Result-Aware query plans. It ensures that every sub-query of the query plan effectively contributes to the results of the query. We propose FedUP, a new federation engine that builds such plans by tracking the provenance of query results. However, getting query results requires computing source selection, while computing source selection requires query results. To break this vicious cycle, FedUP computes results and provenances on tiny quotient summaries [4] of federations, but at the cost of the source selection accuracy. The contributions of this paper are the following:

- We define and formalize the concept of Result-Aware query plans. Any federation query optimizer can safely use such a query plan. The overall idea is to normalize the logical plan and prune subexpressions that do not contribute to the final results of the query.
- We describe an algorithm that computes Result-Aware query plans. The proof of the correctness of the algorithm is detailed in the appendix.
- We present how we compute quotient summaries and how Result-Aware query plans can be effectively obtained by running our algorithm on such summaries.
- We evaluate FedUP on LargeRDFBench [21] and FedShop [9]. Our experiments empirically demonstrate that: (i) FedUP is on par with state-of-the-art federation engines [20, 22] on small federations. (ii) FedUP drastically outperforms state-of-the-art federation engines on large federations of SPARQL endpoints.

This paper is organized as follows: Section 2 presents the background and motivations. Section 3 defines the Result-Aware source selection problem and presents our solution to this problem. Section 4 presents our experimental results conducted on federations of endpoints. Section 5 reviews related work about federation engines. Section 6 concludes and outlines future work.

---

[1]We stopped the execution after 1 hour.

## 2 BACKGROUND AND MOTIVATIONS

We assume that the reader is familiar with the concepts of RDF and core SPARQL [16, 23], i.e., triple patterns (tp), basic graph patterns (BGP), AND, UNION, FILTER, and OPTIONAL graph patterns.

*Definition 2.1 (SPARQL Federation [7, 12]).* A SPARQL federation $F$ is a set of federation members $(G, I_{sparql})$ where $G$ is an RDF graph and $I_{sparql}$ is a SPARQL endpoint interface to access $G$.

*Definition 2.2 (Federated Query Evaluation [7]).* The evaluation $\llbracket Q \rrbracket_F$ of a federated query $Q$ over a federation $F$ is a set of solutions mappings defined as $\llbracket Q \rrbracket_F = \llbracket Q \rrbracket_{G_{union}}$ where $G_{union} = \cup_{(G,I) \in F} G$.

One major challenge for federation engines consists in solving the *source selection problem*, i.e., finding the minimal set of federation members to contact for each triple pattern of the query.

PROBLEM 1 (SOURCE SELECTION [19]). *Given a SPARQL query $Q$ and a federation $F$, find the minimal set of federation members $R(tp) \subseteq F$ for each triple pattern $tp \in Q$ where $\forall (G,I) \in R(tp), \exists \mu \in \llbracket Q \rrbracket_F$ such that $\mu(tp) \in G$.*

The result of source selection can be represented as FedQPL expressions [7, 8]. FedQPL is a language to represent logical query plans over heterogeneous federations.

*Definition 2.3 (FedQPL expression [7, 8]).* A FedQPL expression is an expression $\varphi$ that can be constructed from the following grammar, in which $req, filter, mj, mu$, and $leftjoin$ are terminal symbols, $tp$ is a triple pattern, $f$ is a federation member, $R$ is a SPARQL filter condition, and $\Phi$ is a non-empty set of FedQPL expressions.

$$\varphi ::= req_f^{tp} \mid filter^R(\varphi) \mid mu\Phi \mid mj\Phi \mid leftjoin(\varphi, \varphi)$$

*Definition 2.4 (FedQPL semantics [7, 8]).* Let $\varphi$ be a FedQPL expression, the solutions mappings obtained with $\varphi$, denoted by $sols(\varphi)$, is a set of solutions mappings that is defined recursively as follows:

(1) If $\varphi$ is of the form $req_f^{tp}$ then
$$sols(\varphi) = \llbracket tp \rrbracket_f$$

(2) If $\varphi$ is of the form $filter^R(\varphi')$ then
$$sols(\varphi') = \{\mu | \mu \in sols(\varphi') \land \mu \vDash R\}$$

(3) If $\varphi$ is of the form $mj\Phi$ where $\Phi = \{\varphi_1, \cdots, \varphi_n\}$ then
$$sols(\varphi) = sols(\varphi_1) \bowtie \cdots \bowtie sols(\varphi_n)$$

(4) If $\varphi$ is of the form $mu\Phi$ where $\Phi = \{\varphi_1, \cdots, \varphi_n\}$ then
$$sols(\varphi) = sols(\varphi_1) \cup \cdots \cup sols(\varphi_n)$$

(5) If $\varphi$ is of the form $leftjoin(\varphi_1, \varphi_2)$ then
$$sols(\varphi) = sols(\varphi_1) ⟕ sols(\varphi_2)$$

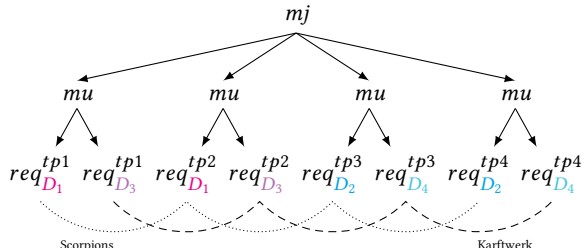

(a) $S6_j$: The joins-over-unions logical plan fails to capture the relationship between bindings.

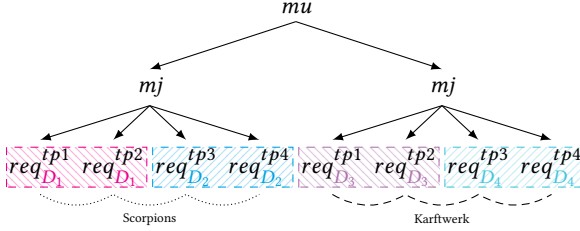

(b) $S6_u$ The unions-over-joins logical plan provides 2 sub-trees accurately capturing the 2 combinations required to create the results.

**Figure 3: Logical plans for query S6 over the $F$ federation expressed in FedQPL Language.**

**Example 1** (Query $S6$ over Federation $F_1$). Consider the federation $F_1$ in Figure 2 and the query $S6$ from FedBench [21]. $S6$ returns the names of artists located near the Federal Republic of Germany:

```
SELECT * WHERE {
    ? artist   foaf:name ?name .                              #tp1 @ D1, D3
    ? artist   foaf:based_near ? location  .                  #tp2 @ D1, D3
    ?location  geo:parentFeature ?germany .                   #tp3 @ D2, D4
    ?germany geo:name "Federal Republic  of  Germany" . }     #tp4 @ D2, D4
```

The evaluation of $S6$ over $F_1$ returns 2 solutions mappings:

|   | ?artist | ?name | ?location | ?germany |
|---|---|---|---|---|
| $\mu_1$ | http://D1/Scorpions | Scorpions | http://D2/Hanover | http://D2/Germany |
| $\mu_2$ | http://D1/Kraftwerk | Kraftwerk | http://D4/Berlin | http://D4/Germany |

**Example 2** (Joins-over-unions logical plans). The minimal source selection of the query $S6$ over the federation $F_1$ is $R(tp1) = \{D_1, D_3\}$, $R(tp2) = \{D_1, D_3\}$, $R(tp3) = \{D_2, D_4\}$, and $R(tp4) = \{D_2, D_4\}$. Such a source selection can be represented as a joins-over-unions FedQPL expression as depicted in Figure 3a:

$$S6_j = mj \{ mu \{req_{D_1}^{tp1}, req_{D_1}^{tp1}\}, mu \{req_{D_1}^{tp2}, req_{D_3}^{tp2}\},$$
$$mu \{req_{D_3}^{tp3}, req_{D_2}^{tp3}\}, mu \{req_{D_2}^{tp4}, req_{D_4}^{tp4}\}\}$$

To evaluate $S6_j$, federation engines must implement the operators and generate at least as many SERVICE queries as the number of $req$ in $S6_j$. Gathering $req$ in exclusive groups constitutes a major performance improvement as it lowers the number of SERVICE queries, pushing more computation on SPARQL endpoints. However, $S6_j$ cannot apply such an optimization.

The main issue with the current definition of source selection is that important information is missing. Based on the results of $S6$, the evaluation of the query $S6$ only requires two series of joins: $\{tp1 \rightarrow D_1, tp2 \rightarrow D_1, tp3 \rightarrow D_2, tp4 \rightarrow D_2\}$ for $\mu_1$, and $\{tp1 \rightarrow D_3, tp2 \rightarrow D_3, tp3 \rightarrow D_4, tp4 \rightarrow D_4\}$ for $\mu_2$. However, with unions ($mu$) under joins ($mj$), this information is hidden from the query optimizer, preventing it from considering other options, and potentially finding better plans. All existing federation engines generate such joins-over-unions plans [7]. By design, they remain blind to many optimizations that their counterpart, unions-over-joins, can perform:

**Example 3** (Unions-over-joins logical plans). Using the results of $S6$ over $F_1$, an alternative to $S6_j$ is the unions-over-joins FedQPL expression $S6_u$ depicted in Figure 3b:

$$S6_u = mu \{mj \{req_{D_1}^{tp1}, req_{D_1}^{tp2}, req_{D_2}^{tp3}, req_{D_2}^{tp4}\},$$
$$mj \{req_{D_3}^{tp1}, req_{D_3}^{tp2}, req_{D_4}^{tp3}, req_{D_4}^{tp4}\}\}$$

Using $S6_u$, federation engines only evaluate joins that actually contribute to the final results of $S6$. Moreover, $S6_u$ allows federation engines to identify more exclusive groups than $S6_j$. For example, triple patterns $tp1$ and $tp2$ are grouped together, as well as $tp3$ and $tp4$:

$$S6'_u = mu \{mj \{req_{D_1}^{tp1,tp2}, req_{D_2}^{tp3,tp4}\}, mj \{req_{D_3}^{tp1,tp2}, req_{D_4}^{tp3,tp4}\}\}$$

In summary, the current source selection definition hides important information about which sources should be combined to find results. With just a set of relevant sources per triple pattern, it is impossible to know which combinations of sources contribute to the final results of queries. Consequently, many valuable query plans such as $S6_u$ remain invisible to the query optimizer. This problem is at the origin of the poor performance of current federation engines on the FedShop benchmark [9]. Solving this problem requires defining a new kind of source selection able to reveal the relevant combination of sources.

## 3 FEDUP: A RESULT-AWARE FEDERATION ENGINE

In this section, we introduce our approach to building a Result-Aware federation engine. We consider a federation $F$ and a core SPARQL query $Q$ composed of BGP patterns with Union, Filters, and Optional. We consider $\varphi$ a valid FedQPL expression for a query $Q$ [7]. In the following, we rely on set-based semantics of SPARQL [16].

As stated in the previous section, existing source selection do not reveal which combinations of relevant sources effectively produce results. Without this information, a class of query plans cannot be explored, such as the union-over-join query plans.

To overcome this problem, a source selection should produce logical plans following a union-over-join grammar where each sub-expression effectively contributes to the final results of the query.

The first step to generate a union-over-join query plan is to rewrite $\varphi$ using equivalence rules.

*Definition 3.1 (Equivalence rules [7, 16]).* Let $\varphi_1, \varphi_2$, and $\varphi_3$ be FedQPL expressions that are valid for $F$. It holds that[2]:

(R1)  join $(\varphi_1, \varphi_2) \overset{F}{\equiv}$ join $(\varphi_2, \varphi_1)$;

(R2)  union $(\varphi_1, \varphi_2) \overset{F}{\equiv}$ union $(\varphi_2, \varphi_1)$;

(R3)  union $(\varphi_1, \varphi_1) \overset{F}{\equiv} \varphi_1$;

(R4)  join $(\varphi_1,$ join $(\varphi_2, \varphi_3)) \overset{F}{\equiv}$ join $($join $(\varphi_1, \varphi_2), \varphi_3)$;

(R5)  union $(\varphi_1,$ union $(\varphi_2, \varphi_3)) \overset{F}{\equiv}$ union $($ union $(\varphi_1, \varphi_2), \varphi_3)$;

(R6)  join $(\varphi_1,$ union $(\varphi_2, \varphi_3)) \overset{F}{\equiv}$ union $($join $(\varphi_1, \varphi_2),$ join $(\varphi_1, \varphi_3))$.

(R7)  leftjoin$($union$(\varphi_1, \varphi_2), \varphi_3) \overset{F}{\equiv}$ union$($leftjoin$(\varphi_1, \varphi_3),$ leftjoin$(\varphi_2, \varphi_3))$

To illustrate, we applied the equivalence rules $[R1 - R7]$ to $S6_j$ of the example 2 and generated $S6'_j$:

$$S6'_j = mu\ \{mj\ \{req^{tp1}_{D_1}, req^{tp2}_{D_1}, req^{tp3}_{D_2}, req^{tp4}_{D_2}\},$$
$$\dots \times 14$$
$$mj\ \{req^{tp1}_{D_3}, req^{tp2}_{D_3}, req^{tp3}_{D_4}, req^{tp4}_{D_4}\}\}$$

We know that only the first and last subexpressions contribute to the results. Consequently, $S6_J$ is not a Result-Aware source selection for the query $S6$. However, if we remove the 14 useless subexpressions, i.e., those returning empty results, we obtain the query plan of $S6_u$ of example 3, a union-over-join query plan with only subexpressions contributing to the final results of $S_6$.

*Definition 3.2 (Result-Aware property).* Let $\varphi$ be a normalized FedQPL expression using [R1-R7] equivalence rules, for a query $Q$ over a federation $F$. $\varphi$ is Result-Aware if:

$$\forall \varphi' \subseteq \varphi, \exists \mu \in sols(\varphi')\ such\ that\ \exists \mu' \in [\![Q]\!]_F, \mu' \subseteq \mu$$

If we only consider SPARQL queries based on conjunctive queries with UNION and FILTER, the normal form of $\varphi$ follows a Union-over-joins grammar $S_{\cup\_(\bowtie)}$ defined as:

$$S_{\cup\_(\bowtie)} = mu\ \{(mj\ \{(req^{tp}_D)^+\})^+\}$$

In the presence of Optional clauses, we need to extend the grammar of the class $S_{\cup\_(\bowtie)}$ to include *leftjoin*:

$$\begin{cases} a_u ::= & a_j \mid mu\ \Phi_u \\ a_j ::= & a_b \mid mj\ \Phi_b \mid leftjoin(a_j, a_u) \\ a_b ::= & req^{tp}_D \end{cases}$$

*Definition 3.3 (Result-Aware source selection Problem).* Given a SPARQL query Q and federation $F$, find a $\varphi$ for $Q$ such that $\varphi$ is Result-Aware.

As a Result-Aware source selection is normalized, then pruned with useless subexpressions, some subexpressions can appear several times in $\varphi$. We do not require to factorize the duplicated subexpressions as we consider that this should be handled by the query optimizer.

**Algorithm 1:** Source Selection $\mathcal{A}$ for a query $Q$ over a federation $F$.

---

**1 Function** $\mathcal{A}(Q, F)$:       ▷ Root of the logical plan

**2**   **return** $mu\ \mathcal{A}'(Q, F)$

**3 Function** $\mathcal{A}'(Q, F)$:    ▷ Explores every graph pattern $Q$

**4**   $\Phi_o \leftarrow \emptyset$

**5**   **if** $Q$ is a triple pattern $tp$ **then**

**6**    $\Phi_o \leftarrow \Phi_o \cup \{req^{tp}_f \mid f \in F\}$

**7**   **else if** $Q$ is $(P_1$ AND $P_2)$ **then**     ▷ $P_1 \bowtie P_2$

**8**    $\Phi_1, \Phi_2 \leftarrow \mathcal{A}'(P_1, F), \mathcal{A}'(P_2, F)$

**9**    $\Phi_o \leftarrow \Phi_o \cup \{mj\{\varphi_1, \varphi_2\} \mid \varphi_1 \in \Phi_1 \wedge \varphi_2 \in \Phi_2\}$

**10**   **else if** $Q$ is $(P_1$ UNION $P_2)$ **then**    ▷ $P_1 \cup P_2$

**11**    $\Phi_1, \Phi_2 \leftarrow \mathcal{A}'(P_1, F), \mathcal{A}'(P_2, F)$

**12**    $\Phi_o \leftarrow \Phi_o \cup \{\varphi \mid \varphi \in \Phi_1 \vee \varphi \in \Phi_2\}$

**13**   **else if** $Q$ is $(P_1$ OPTIONAL $P_2)$ **then**   ▷ $P_1 ⟕ P_2$

**14**    $\Phi_1, \Phi_2 \leftarrow \mathcal{A}'(P_1, F), \mathcal{A}'(P_2, F)$

**15**    **for** $\varphi_1 \in \Phi_1$ **do**

**16**     $\Phi^{\varphi_1}_{join} \leftarrow \{\varphi_2 \mid \varphi_2 \in \Phi_2 \wedge sols(mj\{\varphi_1, \varphi_2\}) \neq \varnothing\}$

**17**     **if** $\Phi^{\varphi_1}_{join} = \varnothing$ **then** $\Phi_o \leftarrow \Phi_o \cup \{\varphi_1\}$

**18**     **else** $\Phi_o \leftarrow \Phi_o \cup \{leftjoin(\varphi_1, mu\ \Phi^{\varphi_1}_{join})\}$

**19**   **else if** $Q$ is $(P$ FILTER $R)$ **then**

**20**    $\Phi \leftarrow \mathcal{A}'(P, F)$

**21**    $\Phi_o \leftarrow \Phi_o \cup \{filter^R(\varphi) \mid \varphi \in \Phi\}$

**22**   **return** $\{\varphi \mid \varphi \in \Phi_o \wedge sols(\varphi) \neq \varnothing\}$

---

## 3.1 Solving the Result-Aware source selection problem

Algorithm 1 builds a Result-Aware source selection for query $Q$ over a federation $F$ based on a set of recursive rules. The algorithm is designed with 2 main ideas:

(1) Build a union-over-join plan.
(2) Just keep expressions that contribute to the final results of the query.

To reach this objective, the algorithm evaluates the query on the federation, extracts provenance from solution mappings, and produces the corresponding FedQPL expression.

To ensure that every produced $\varphi$ expression is based on results, we rely on $sols(\varphi)$ as a function that returns the mappings resulting in the evaluation of the expression $\varphi$ over $F$. Proofs of correctness, completeness and result-awareness are available in appendix A.2.

We illustrate this algorithm on Query $S6$ of the example 2 over Federation $F_1$ of Figure 2. First, the algorithm merges all upcoming subexpressions with a multi-union at Line 1. Then, it enters Line 7 with $(tp1$ AND $(tp2$ AND $(tp3$ AND $tp4)))$. Line 3 states that evaluating $tp3$ and $tp4$ both returns $\{req^{tp}_{D_2}, req^{tp}_{D_4}\}$. Then, Line 22 checks that their intersections return mappings. Here, $mj\ \{req^{tp3}_{D_2}, req^{tp4}_{D_2}\}$ and $mj\ \{req^{tp3}_{D_4}, req^{tp4}_{D_4}\}$ indeed return mappings, but most importantly:

$$mj\ \{req^{tp3}_{D_2}, req^{tp4}_{D_4}\} = \varnothing$$
$$mj\ \{req^{tp3}_{D_4}, req^{tp4}_{D_2}\} = \varnothing$$

---

[2] $leftjoin(\varphi_1, union(\varphi_2, \varphi_3)) \overset{F}{\equiv} union(leftjoin(\varphi_1, \varphi_2), leftjoin(\varphi_1, \varphi_3))$ does not hold. See counter example in Appendix A.1.

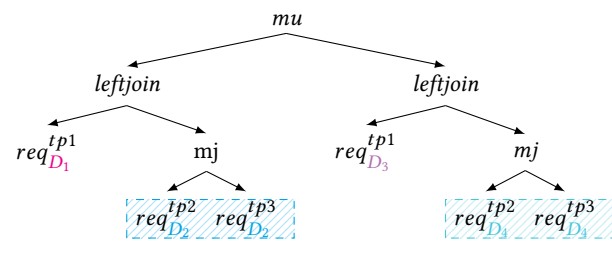

**Figure 4: Logical plan for Query $S7$ with `OPTIONAL`.**

Only the former expressions are kept, the latter ones are discarded. After applying the joining rule for every triple pattern and simplifying nested multi-join expressions, we obtain Figure 3b's expected plan:

$$mu \{ mj \{ req_{D_1}^{tp1}, req_{D_1}^{tp2}, req_{D_2}^{tp3}, req_{D_2}^{tp4} \},$$
$$mj \{ req_{D_3}^{tp1}, req_{D_3}^{tp2}, req_{D_4}^{tp3}, req_{D_4}^{tp4} \} \}$$

The 4 exclusive groups are easily identified: `tp1.tp2` at $D_1$ and $D_3$, `tp3.tp4` at $D_2$ and $D_4$.

**Example 4** (Optional Query $S7$ over Federation $F_1$). To illustrate Result-Aware query plan in the presence of `OPTIONAL`, we consider the query $S7$:

```
SELECT * WHERE {
  ? artist  foaf:based_near ?location  .   #tp1
  OPTIONAL {
    ?location  geo:parentFeature ?germany .  #tp2
    ?germany geo:name "Federal Republic of Germany" . }}  #tp3
```

For such a query $S7$, and as depicted in Figure 4, Algorithm 1 produces a multi-union of 2 left joins:

$$S7_u = mu \{ leftjoin(req_{D_1}^{tp1}, mu \{ mj \{ req_{D_2}^{tp2}, req_{D_2}^{tp3} \} \})$$
$$leftjoin(req_{D_3}^{tp1}, mu \{ mj \{ req_{D_4}^{tp2}, req_{D_4}^{tp3} \} \}) \}$$

Thanks to the line 16, `OPTIONAL` is ensured to be Result-Aware. The evaluation of $S7_u$ on $F_1$ returns the expected results.

## 3.2 FedUP on summaries

FedUP introduces a vicious cycle: our source selection requires query results, and query results require computing source selection. To tackle this issue, we execute Algorithm 1 on a tiny quotient summary [4, 5] of the federation.

*Definition 3.4 (Quotient RDF summary [5]).* Given an RDF graph $G$ and an RDF node equivalence relation $\psi$, the summary of $G$ by $\psi$, which is an RDF graph denoted $\psi(G)$, is the quotient of $G$ by $\psi$.

Quotient summaries have many interesting properties that are relevant in the context of the source selection problem. First, queries that have answers on $F$ also have answers on $\psi(F)$, enabling FedUP to ensure complete results. Abusing notation, $\psi(F)$ is the quotient summary of $F$, i.e., the federation obtained by replacing all RDF graphs $G$ in $F$ by the quotient summary of $G$. Second, quotient summaries are RDF graphs. The source selection algorithm is the same

whether it is executed over the federation or a quotient summary of the federation. Finally, quotient summaries preserve edges in graphs.

*Definition 3.5 (Summary representativeness [5]).* Given a SPARQL query $Q$, a federation $F$, and an RDF node equivalence relation $\psi$, if $[\![Q]\!]_F \neq \emptyset$ then we have $[\![\psi(Q)]\!]_{\psi(F)} \neq \emptyset$.

FedUP uses $\psi_h$ as the RDF node equivalence relation to summarize SPARQL federations. It is defined as follows:

$$\psi_h(node) = \begin{cases} authority(node) & \text{if node is an IRI} \\ \text{"any"} & \text{if node is a Literal} \end{cases}$$

$\psi_h$ is based on the HiBISCuS summary [19], and replaces IRIs by their authority and Literals by "lit". To illustrate, the quotient of the federation $F_1$ by $\psi_h$ is a federation $\psi_h(F_1)$ comprising 8 quads:

| http://D1 | foaf:based_near | http://D2 | http://D1 |
| http://D1 | foaf:name | "any" | http://D1 |
| http://D2 | geo:parentFeature | http://D2 | http://D2 |
| http://D2 | geo:names | "any" | http://D2 |
| http://D3 | foaf:based_near | http://D4 | http://D3 |
| http://D3 | foaf:name | "any" | http://D3 |
| http://D4 | geo:parentFeature | http://D4 | http://D4 |
| http://D4 | geo:names | "any" | http://D4 |

On this simple example, both $F_1$ and $\psi_h(F_1)$ have the same size. However, in practice, $\psi_h$ generates summaries that are orders of magnitude smaller than original federations as shown in Figure 2. Although very compact, experimental results demonstrate that quotient summaries generated by $\psi_h$ allows FedUP to find efficient query plans. The intuition behind $\psi_h$ is that authorities alone allow federation engines to identify which endpoints host a specific triple.

To build Result-Aware source selection, FedUP applies the same summary function $\psi_h$ to triple patterns of the input query. As $\psi_h$ projects all literals on one constant, most query filters cannot be properly evaluated and are removed. Our motivating query $S6$ remains identical, except for the literal "Federal Republic of Germany" that becomes "any". As the Result-Aware property is now ensured on the summary and not on the original federation, some subexpressions of the query plan may return empty results on the federation.

To illustrate, applying the Algorithm 1 on the summary graph $\psi_h(F)$ with the modified query $S6$ returns a multi-union of 12 multi-joins instead of 2 on $F$. However, experimental results demonstrate that query plans generated using summaries remain very efficient.

## 4 EXPERIMENTAL STUDY

This experimental study aims to empirically answer the following questions:

(1) Does FedUP perform better than existing engines on LargeRDF-Bench?
(2) Does FedUP perform better than existing engines when the size of the federation grows?

To conduct the experiment study, we implemented FedUP on top of FedX [22]. FedUP produces Result-Aware source selection plans that are optimized and executed by FedX. Similarly to many

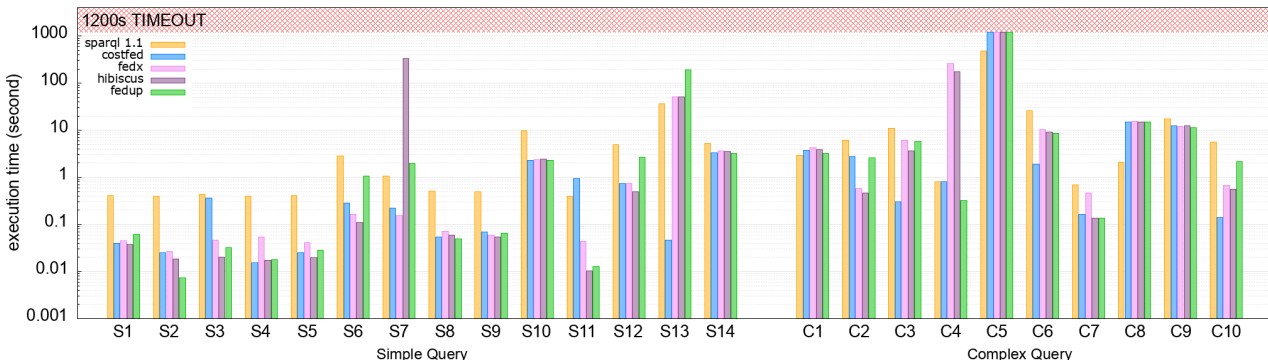

**Figure 5: Query execution time achieved by federation engines for simple and complex queries of LargeRDFBench.**

**Table 2: The size of the summaries of the federation engines.**

|  | FedShop20 | FedShop200 | LargeRDFBench |
|---|---|---|---|
| **Federation** | 5,167,810 quads | 41,821,489 quads | 1,004,491,996 quads |
| **FedUP** | 76KB (0.6K quads) | 767KB (1K quads) | 705KB (6K quads) |
| **CostFed** | 9MB | 95MB | 11MB |
| **HiBISCuS** | 892KB | 9MB | 539KB |
| **SemaGrow** | 1,8MB | 18MB | |

state-of-the-art federation engines [20, 22], FedUP performs an additional pruning step by performing ASK queries in the presence of general predicate with constant. The summary and ASK queries allow FedUP to build accurate logical plans even in large-scale federations. Code, configurations, queries, and datasets are available on the GitHub platform at https://anonymous.4open.science/r/FedUP-experiments-7F21.

### 4.1 Experimental Setup

We used 2 benchmarks:

(1) LargeRDFBench [18] is the most commonly used benchmark to evaluate the performance of federation engines [1, 6, 19, 20, 22]. The benchmark is explicitly designed to represent federated SPARQL queries on real-world datasets. In our experiments, the workload comprises 14 simple queries (S) and 10 complex queries (C). Each dataset is loaded into a separate endpoint, resulting in a total of 14 endpoints. These queries cover all types of core SPARQL operators such as UNION, OPTIONAL, and FILTER. However, LargeRDFBench cannot scale on the number of sources.

(2) FedShop [9] is a new Benchmark allowing to scale on the number of sources. FedShop [9] is designed to study the scalability of federation engines in terms of the number of endpoints. It provides queries and datasets from 20 endpoints up to 200 endpoints. The queries cover all types of core SPARQL operators. The query workload consists of 10 instances for each of the 12 templates, resulting in 120 instances. Each instance is generated by replacing placeholders in the template with randomly selected values. Query templates are organized into 3 levels of source selection

difficulties: Single-Domain (SD), Multi-Domain (MD), and Cross-Domain (CD). SD queries are restricted to a single source with no global join variables and bound triple pattern subjects. MD queries can be assessed on multiple sources without global join variables and unbound subjects. CD queries can be broken down into subqueries, each evaluated on different sources, requiring global join variables.

To run FedUP, we computed the quotient summaries for LargeRDFBench and FedShop. Table 2 represents the size of the summaries for each federation engine. The summaries of FedUP remains very compact, although multiplying the size of the federation by 10 increases the size of FedUP's summary by a multiplicative factor of 10 in FedShop.

For the two experiments, all federation graphs are stored as named graphs in a single Virtuoso endpoint (Version 7.2.7.3234-pthreads).

To run our experiments, we used a local cloud instance with Ubuntu 20.04.4 LTS, a AMD EPYC 7513-Core processor with 16 vCPUs allocated to the VM, 1TB SSD, and 64GB of RAM. The Virtuoso endpoint hosting the data, as well as the federation engines, ran on the same machine.

### 4.2 LargeRDFBench: Parity among engines

In this experiment, we compared FedUP with FedX [22], HiBISCuS [19] (Ask dominant), and CostFed [20] (Ask dominant). We also included the SPARQL 1.1 Service queries available in the LargeRDFBench that we executed with Apache Jena. These queries are hand-crafted with predefined source selection.

Figure 5 presents the performance of the different engines. The x-axis represents the competitors for each query, and the y-axis displays the execution time on a logarithmic scale. This execution time is defined as the time spent by each federation engine from source selection to federated query execution. Each query underwent 5 runs, and the reported measurements in Figure 5 are the averages of these runs. We set a 20-minute timeout before stopping the federated query execution.

The x-axis represents the competitors for each query, and the y-axis displays the execution time on a logarithmic scale. This execution time is defined as the time spent by each federation engine from source selection to federated query execution. Each

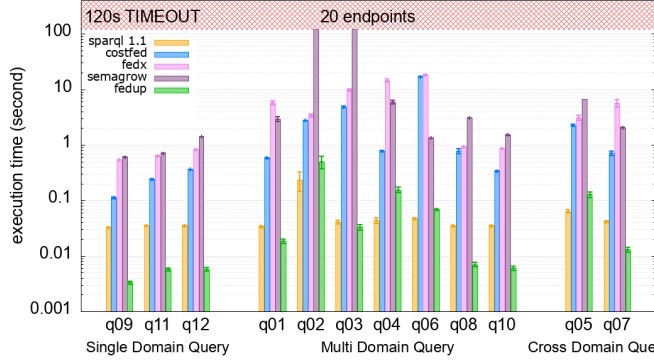
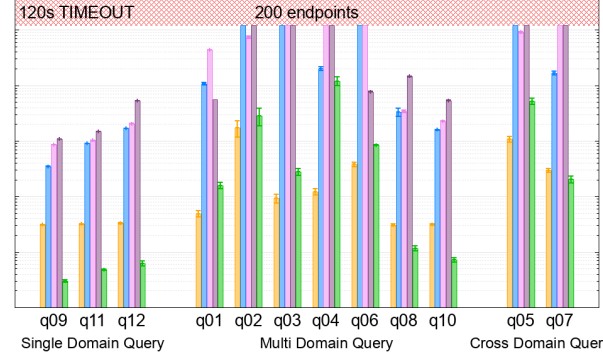

**Figure 6: Query execution time of federation engines on FedShop queries for 20 and 200 endpoints.**

query underwent 5 runs, and the reported measurements in Figure 5 are the averages of these runs. We set a 20-minute timeout before stopping the federated query execution.

Figure 5 shows that most of the time, federation engines yield comparable execution time. Federation engines build logical plans that are equivalent. This is attributed to either the simplicity of queries (20 out of 24 queries require a single combination of sources), or their low selectivity regarding source selection for FedUP to make a difference.

Figure 5 shows that, most notably on Query $S13$, CostFed outperforms its competitors by order of magnitudes, which highlights the need for better join ordering: when disabled, CostFed executes Query $S13$ in $10s$ instead of $50ms$.

Overall, FedUP does not provide significant improvements over state-of-the-art on LargeRDFBench. In the LargeRDFBench context, Join-over-Unions query plans are also Result-Aware query plans.

## 4.3 FedShop: FedUP outperforms other engines

In this experiment, we compared FedUP with FedX [22], Semagrow [6], and CostFed [20] (Ask Dominant). Fedshop comes with RSA queries written as SPARQL 1.1 Service queries that we executed with Apache Jena. These queries are hand-crafted with predefined source selection that follows union-over-joins logical plans.

Figure 6 reports the performance of federation engines in terms of execution time. We run the 10 configurations of FedShop but only reported results for the 20 and 200 endpoints configurations. The x-axis denotes the query templates.

For each templated query, each bar on the x-axis represents the evaluated engine, while its height represents the average execution time of the templated queries on a logarithmic scale. On the left, the federation comprises 20 endpoints, while on the right, the federation comprises 200 endpoints. The timeout is configured for 120 seconds to align with FedShop's focus on interactive eCommerce use-cases, where end-users anticipate receiving results quickly.

Figure 6 shows that, for all queries, FedUP outperforms its competitors from 1 to 3 orders of magnitude in terms of execution time. On SD queries, FedUP's summary allows it to efficiently find the best logical plan comprising a single exclusive group. FedUP built its quotient summary using the authority of URIs, and since SD queries stay on a single domain, evaluating the source selection

query on this summary is fast and accurate. Competitors find the same plan but spend most of the time in source selection. For instance, CostFed spends $1s$ of source selection for $10ms$ of actual execution on template $q12$ when there are 200 endpoints.

On MD queries, FedUP remains close to the baseline except for $q04$ with 200 endpoints. Similarly to SD queries, FedUP's summary allows it to efficiently find the minimal set of combinations, each comprising a single exclusive group. The baseline and FedUP prove that a federation engine could execute these queries under $2s$, however, competing engines present drastically worse execution times, even reaching the 2-minute timeout on occasions when 200 endpoints are involved. Their source selection process is fast but builds joins-over-unions plans that cannot be transformed into efficient physical plans: not only do they fail to identify exclusive groups, but they create combinations without results that still need to be checked at execution time, hence wasting resources. For $q04$, FedUP and CostFed build equivalent plans that need to check numerous combinations without results, hence providing similar performance.

On CD queries, FedUP remains close to the baseline as well. It extensively uses ASK queries to kickstart its source selection query execution, restricting the research space of solutions mappings to build its logical plans. For $q07$, its plans are equivalent to the SPARQL 1.1 baseline. However, for $q05$, FedUP creates plans of 200 combinations while the baseline needs 17 combinations on average, hence spending more time to evaluate the federated query.

Figure 6 shows that half of the time, FedUP performs better than the RSA SPARQL 1.1 queries. Indeed, FedUP benefits from parallel execution, where up to 8 FedX instances are in charge of executing subparts of the logical plan. The baseline uses Apache Jena to evaluate its SERVICE queries and, therefore, does not benefit from such a feature.

Overall, FedUP outperforms state-of-the-art federations engines by order of magnitudes. Thanks to its summary and ASK queries, FedUP quickly produces better logical plans. Consequently, FedUP can execute the federated query before reaching the timeout even on large-scale federations comprising up to 200 endpoints.

## 5 RELATED WORK

Given a federation of SPARQL endpoints, federation engines process SPARQL queries in three steps[2]: (i) Source selection and

query decomposition, (ii) query optimization, and (iii) query execution.

Currently, the first step produces a logical plan[7] with a set of sources to contact per triple pattern [20]. Some engines suppose the existence of summaries computed over the federation of SPARQL endpoints [3, 6, 10, 10, 11, 14, 15, 17, 19, 20, 25]. Others are Zero-knowledge, FedX or Lusail [1, 22] just require a catalog of SPARQL endpoints.

To minimize the number of relevant sources per triple pattern, some engines such as [22] perform triple pattern-aware source selection, i.e., they send ask queries on endpoints to be sure that a triple pattern return at least one result. Other engines such as [19, 20] perform BGP-aware source selection, i.e., they detect and prune sources that do not contribute to the final results of the query. However, BGP-aware source selection produces a join-over-union plan that may not be Result-Aware.

Once the source selection is established if several triple patterns share the same single source, it is possible to group them into exclusive groups as proposed by FedX [22]. Exclusive groups are more likely to happen with Result-Aware query plans, as shown in Figure 3b compared to Figure 3a

Lusail [1] improved the grouping of sources by determining if join variables are local or global using set-differences. Such technique is zero-knowledge as set-difference is computed online using simple filter-not-exist queries. The Lusail grouping approach can be applied on top of existing source selection techniques, including Result-Aware source selection.

## 6 CONCLUSION

In this paper, we introduced the new concept of Result-Aware source selection . Result-Aware query plans ensure that all combinations of relevant sources contribute to the final results of the query.

Building a Result-Aware query plan is driven by results; however, query results are unavailable when computing source selection. We solved this issue by computing a Result-Aware query plan on quotient summaries. Of course, summaries introduce inaccuracies; however, the results of benchmarks demonstrate huge performance improvements, especially when the size of the federation grows. On the FedShop Benchmark, Result-Aware query plan outperforms traditional approaches by one order of magnitude, offering new perspectives for federated query processing.

In future work, we plan to support the MINUS/Filter, not EXISTS, in SPARQL queries. There is also important room for improvement for optimizing unions-over-joins query plan. $\varphi$ expression can be factorized, and join order can be improved to fill the remaining gap with RSA queries in the FedShop benchmark.

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

# A APPENDIX

## A.1 Equivalence Rules

Let $\varphi_1, \varphi_2$, and $\varphi_3$ be FedQPL expressions that are valid for $F$. In a federation context, it does not holds that:

(R8)  $leftjoin(\varphi_1, union(\varphi_2, \varphi_3)) \overset{F}{\equiv} union(leftjoin(\varphi_1, \varphi_2), leftjoin(\varphi_1, \varphi_3))$

To illustrate, let us consider the query $Q_o$ with 1 OPTIONAL and 2 triple patterns:

```
SELECT * WHERE {
    ?artist foaf:based_near ?location .   #tp1
    OPTIONAL { ?location geo:parentFeature ?germany . }}  #tp2
```

The federation $F$ comprises 2 members $f_1$ and $f_2$ with 3 triples as follows:

| http://f1/Scorpions | foaf:based_near | http://f1/Hanover | http://f1 |
| http://f1/Kraftwerk | foaf:based_near | http://f2/Berlin | http://f1 |
| http://f2/Berlin | geo:parentFeature | http://f2/Germany | http://f2 |

With such a federation $F$ and query $Q_o$, the FedQPL expression is:

$$\varphi_o = leftjoin(req_{f_1}^{tp1}, mu\,\{req_{f_1}^{tp2}, req_{f_2}^{tp2}\})$$

The evaluation $[\![\varphi_o]\!]_F$ of $\varphi_o$ over $F$ returns:

|  | ?artist | ?location | ?germany |
|---|---|---|---|
| $\mu_1$ | http://f1/Scorpions | http://f1/Hanover | |
| $\mu_2$ | http://f1/Kraftwerk | http://f2/Berlin | http://f2/Germany |

However, after applying the equivalence rule $R8$ on $\varphi_o$, we get the following expression $\varphi_{o'}$:

$$\varphi_{o'} = mu\,\{leftjoin(req_{f_1}^{tp1}, req_{f_1}^{tp2}), leftjoin(req_{f_1}^{tp1}, req_{f_2}^{tp2})\}$$

The evaluation of $\varphi_{o'}$ over $F$ returns unexpected results:

|  | ?artist | ?location | ?germany |
|---|---|---|---|
| $\mu_1$ | http://f1/Scorpions | http://f1/Hanover | |
| $\mu_2$ | http://f1/Kraftwerk | http://f2/Berlin | http://f2/Germany |
| $\mu_3$ | http://f1/Kraftwerk | http://f2/Berlin | |

## A.2 $\mathcal{A}$ returns complete and correct results

PROOF. (Completeness) Let $Q$ be a SPARQL query and $F$ be a federation, $\mathcal{A}(Q, F)$ returns complete results if and only if $\forall \mu \in [\![Q]\!]_F, \mu \in sols(\mathcal{A}(Q, F))$. To demonstrate that $\mathcal{A}$ returns complete results, we use the FedQPL equivalences with the SPARQL algebra as defined in Definition 6 [7]. We proceed by contradiction assuming that $\exists \mu \notin sols(\mathcal{A}(Q, F)), \mu \in [\![Q]\!]_F$.

- If $Q$ is a triple pattern $tp$ then $\mu \notin sols(\mathcal{A}(Q, F))$
  $\Leftrightarrow \mu \notin \bigcup_{f \in F} sols(req_f^{tp})$
  $\Leftrightarrow \mu \notin \bigcup_{f \in F} [\![tp]\!]_f$
  $\Leftrightarrow \mu \notin [\![Q]\!]_F$
- If $Q$ is $P_1$ AND $P_2$ then $\mu \notin sols(\mathcal{A}(Q, F))$
  $\Leftrightarrow \mu \notin \bigcup_{\varphi_1, \varphi_2 \in \Phi_1, \Phi_2} sols(mj\{\varphi_1, \varphi_2\})$
  $\Leftrightarrow \mu \notin (\bigcup_{\varphi_1 \in \Phi_1} sols(\varphi_1)) \bowtie (\bigcup_{\varphi_2 \in \Phi_2} sols(\varphi_2))$
  $\Leftrightarrow \mu \notin [\![P_1]\!]_F \bowtie [\![P_2]\!]_F$
  $\Leftrightarrow \mu \notin [\![Q]\!]_F$
- If $Q$ is $P_1$ UNION $P_2$ then $\mu \notin sols(\mathcal{A}(Q, F))$
  $\Leftrightarrow \mu \notin (\bigcup_{\varphi_1 \in \Phi_1} sols(\varphi_1)) \cup (\bigcup_{\varphi_2 \in \Phi_2} sols(\varphi_2))$
  $\Leftrightarrow \mu \notin [\![P_1]\!]_F \cup [\![P_2]\!]_F$
  $\Leftrightarrow \mu \notin [\![Q]\!]_F$

- If $Q$ is $P_1$ FILTER $R$ then $\mu \notin sols(\mathcal{A}(Q, F))$
  $\Leftrightarrow \mu \notin \bigcup_{\varphi \in \Phi} sols(filter^R(\varphi))$
  $\Leftrightarrow \mu \notin \{\mu' \mid \mu' \in \bigcup_{\varphi \in \Phi} sols(\varphi) \wedge \mu' \vDash R\}$
  $\Leftrightarrow \mu \notin \{\mu' \mid \mu' \in [\![P_1]\!]_F \wedge \mu' \vDash R\}$
  $\Leftrightarrow \mu \notin [\![Q]\!]_F$
- If $Q$ is $P_1$ OPTIONAL $P_2$ then $\mu \notin sols(\mathcal{A}(Q, F))$
  $\Leftrightarrow \mu \notin \bigcup_{\varphi_1 \in \Phi_1} leftjoin(\varphi_1, \Phi_{join}^{\varphi_1})$
  $\Leftrightarrow \mu \notin (X \cup ((\bigcup_{\varphi_1 \in \Phi_1} sols(\varphi_1)) \setminus X))$
  $\Leftrightarrow \mu \notin (Y \cup ((\bigcup_{\varphi_1 \in \Phi_1} sols(\varphi_1)) \setminus Y))$
  $\Leftrightarrow \mu \notin ([\![P_1 \bowtie P_2]\!]_F \cup ([\![P_1]\!]_F \setminus [\![P_1 \bowtie P_2]\!]_F))$
  $\Leftrightarrow \mu \notin [\![Q]\!]_F$
  $\Phi_{join}^{\varphi_1} = \{\varphi_2 \mid \varphi_2 \in \Phi_2 \wedge sols(mj\{\varphi_1, \varphi_2\}) \neq \varnothing\}$
  $X = \bigcup_{\varphi_1 \in \Phi_1} sols(mj\{\varphi_1, mu\{\Phi_{join}^{\varphi_1}\}\})$
  $Y = \bigcup_{\varphi_1 \in \Phi_1} sols(mj\{\varphi_1, mu\{\varphi_2 \mid \varphi_2 \in \Phi_2\}\})$
  $\qquad\qquad\qquad\qquad\qquad\qquad\qquad\qquad\qquad\qquad\square$

PROOF. (Correctness) Let $Q$ be a SPARQL query and $F$ be a federation, $\mathcal{A}(Q, F)$ returns correct results if and only if $\forall \mu \in \mathcal{A}(Q, F), \mu \in [\![Q]\!]_F$. The proof of correctness is analogous to the proof of completeness. $\qquad\qquad\qquad\qquad\square$

## A.3 $\mathcal{A}$ returns Result-Aware FedQPL expressions

PROOF. Let $Q$ be a SPARQL query and $F$ be a federation such that $[\![Q]\!]_F \neq \varnothing$. Let $\varphi = \mathcal{A}(Q, F)$ be a FedQPL expression that is not Result-Aware. Consequently, it exists $\varphi' \subseteq \varphi$ such that $\varphi'$ does not contribute to $[\![Q]\!]_F$.

- If $Q$ is a triple pattern $tp$, $\varphi$ is not Result-Aware if it exists $\varphi'$ in $\Phi_{TP}$ such that $sols(\varphi') = \varnothing$, which is impossible by definition of $\Phi_{TP}$. Consequently, $\varphi$ is Result-Aware.
- If $Q$ is $P_1$ AND $P_2$, $\varphi$ is not Result-Aware if
  (1) it exists $\varphi'$ in $\Phi_{JOIN}$ such that $sols(\varphi') = \varnothing$, which is impossible by definition of $\Phi_{JOIN}$.
  (2) it exists $\varphi_1$ in $\Phi_1$ such that $\varphi_1$ does not contribute to $[\![Q]\!]_F$. If $\varphi_1 \subset \varphi$, it exists $\varphi_{join} = mj\{\varphi_1, \varphi_2\}$ in $\Phi_{JOIN}$. By definition, if $sols(\varphi_{join}) \neq \varnothing$, both $\varphi_1$ and $\varphi_2$ contribute to $sols(\varphi_{join})$. As $\varphi_{join}$ contributes to $[\![Q]\!]_F$, $\varphi_1$ also contributes to $[\![Q]\!]_F$.
  (3) it exists $\varphi_2$ in $\Phi_2$ such that $\varphi_2$ does not contribute to $[\![Q]\!]_F$. For the same reason as $\varphi_1$, if $\varphi_2 \subset \varphi$, $\varphi_2$ contributes to $[\![Q]\!]_F$.
  (4) it exists $\varphi' \subset \varphi_1$ where $\varphi_1 \in \Phi_1$ such that $\varphi_1$ contributes to $[\![Q]\!]_F$ but $\varphi'$ does not. By induction, we assume that $\mathcal{A}(P_1, F)$ generates a Result-Aware FedQPL expression. As $\mathcal{A}(P_1, F) = mu\Phi_1$, all FedQPL expressions and subexpressions in $\Phi_1$ contribute to $[\![P_1]\!]_F$. As $\varphi_1$ contributes to $[\![Q]\!]_F$, all subexpressions $\varphi' \subset \varphi_1$ also contributes to $[\![Q]\!]_F$.
  (5) it exists $\varphi' \subset \varphi_2$ where $\varphi_2 \in \Phi_2$ such that $\varphi_2$ contributes to $[\![Q]\!]_F$, but $\varphi'$ does not. Using the same reasoning as for $\varphi' \subset \varphi_1$, we demonstrate that all subexpressions $\varphi' \subset \varphi_2$ contributes to $[\![Q]\!]_F$.

  As a result, if $Q$ is $P_1$ AND $P_2$, it does not exist $\varphi' \subset \varphi$ such that $\varphi'$ does not contribute to $[\![Q]\!]_F$, consequently, $\varphi$ is Result-Aware.
- If $Q$ is $P_1$ OPTIONAL $P_2$, $\varphi$ is not Result-Aware if

(1) it exists $\varphi'$ in $\Phi_{OPT}$ such that $sols(\varphi') = \varnothing$, which is impossible by definition of $\Phi_{OPT}$.

(2) it exists $\varphi_1 \in \Phi_1$ such that $\varphi_1$ does not contribute to $[\![Q]\!]_F$. If $\varphi_1 \subset \varphi$, there are two cases: (a) $\varphi_1 \in \Phi_{OPT} \setminus \Phi_1$. In this case, it exists $\varphi_{opt} = leftjoin(\varphi_1, mu\Phi_{join}^{\varphi_1})$ in $\Phi_{OPT}$. By definition, if $sols(\varphi_{opt}) \neq \varnothing$, $\varphi_1$ contributes to $[\![Q]\!]_F$. (b) $\varphi_1 \in \Phi_{OPT} \cap \Phi_1$. In this case, $\varphi_1$ contributes to $[\![Q]\!]_F$ by definition of $\Phi_{OPT}$.

(3) it exists $\varphi_2 \in \Phi_2$ such that $\varphi_2$ does not contribute to $[\![Q]\!]_F$. If $\varphi_2 \subset \varphi$, it exists $\varphi_{opt} = leftjoin(\varphi_1, mu\Phi_{join}^{\varphi_1})$ in $\Phi_{OPT}$ such that $\varphi_2 \in \Phi_{join}^{\varphi_1}$. By definition, if $\varphi_2 \in \Phi_{join}^{\varphi_1}$ then $sols(mj\{\varphi_1, \varphi_2\}) \neq \varnothing$, and $\varphi_2$ contributes to $sols(\varphi_{opt})$. Consequently, $\varphi_2$ contributes to $[\![Q]\!]_F$.

(4) it exists $\varphi' \subset \varphi_1$ where $\varphi_1 \in \Phi_1$ such that $\varphi_1$ contributes to $[\![Q]\!]_F$ but $\varphi'$ does not. Using the same reasoning as for $\varphi' \subset \varphi_1$ when $Q$ is $P_1$ AND $P_2$, we demonstrate that all subexpressions $\varphi' \subset \varphi_2$ contributes to $[\![Q]\!]_F$ when $Q$ is $P_1$ OPTIONAL $P_2$.

(5) it exists $\varphi' \subset \varphi_2$ where $\varphi_2 \in \Phi_2$ such that $\varphi_2$ contributes to $[\![Q]\!]_F$ but $\varphi'$ does not. We use the same reasoning as for $\varphi' \subset \varphi_1$.

As a result, if $Q$ is $P_1$ OPTIONAL $P_2$, it does not exist $\varphi' \subset \varphi$ such that $\varphi'$ does not contribute to $[\![Q]\!]_F$, consequently, $\varphi$ is Result-Aware.

- If $Q$ is $P_1$ FILTER $R$, $\varphi$ is not Result-Aware if

(1) it exists $\varphi'$ in $\Phi_{FILTER}$ such that $sols(\varphi') = \varnothing$, which is impossible by definition of $\Phi_{FILTER}$.

(2) it exists $\varphi'$ in $\Phi$ such that $\varphi'$ does not contribute to $[\![Q]\!]_F$. If $\varphi' \subset \varphi$, it exists $\varphi_{filter} = filter^R(\varphi')$ in $\Phi_{FILTER}$. By definition, if $sols(\varphi_{filter}) \neq \varnothing$, $\varphi'$ contributes to $sols(\varphi_{filter})$. Consequently, $\varphi'$ contributes to $[\![Q]\!]_F$.

(3) it exists $\varphi'' \subset \varphi'$ where $\varphi' \in \Phi$ such that $\varphi'$ contributes to $[\![Q]\!]_F$ but $\varphi''$ does not. Using the same reasoning as for $\varphi' \subset \varphi_1$ when $Q$ is $P_1$ AND $P_2$, we demonstrate that all subexpressions $\varphi'' \subset \varphi'$ contributes to $[\![Q]\!]_F$ when $Q$ is $P$ FILTER $R$.

As a result, if $Q$ is $P$ FILTER $R$, it does not exist $\varphi' \subset \varphi$ such that $\varphi'$ does not contribute to $[\![Q]\!]_F$, consequently, $\varphi$ is Result-Aware.

- If $Q$ is $P_1$ UNION $P_2$, $\varphi$ is not Result-Aware if

(1) it exists $\varphi'$ in $\Phi_{UNION}$ such that $sols(\varphi') = \varnothing$, which is impossible by definition of $\Phi_{UNION}$.

(2) it exists $\varphi' \subset \varphi_1$ where $\varphi_1 \in \Phi_1$ such that $\varphi_1$ contributes to $[\![Q]\!]_F$ but $\varphi'$ does not. Using the same reasoning as for $\varphi' \subset \varphi_1$ when $Q$ is $P_1$ AND $P_2$, we demonstrate that all subexpressions $\varphi' \subset \varphi_2$ contributes to $[\![Q]\!]_F$ when $Q$ is $P_1$ UNION $P_2$.

(3) it exists $\varphi' \subset \varphi_2$ where $\varphi_2 \in \Phi_2$ such that $\varphi_2$ contributes to $[\![Q]\!]_F$ but $\varphi'$ does not. We use the same reasoning as for $\varphi' \subset \varphi_1$.

As a result, if $Q$ is $P_1$ UNION $P_2$, it does not exist $\varphi' \subset \varphi$ such that $\varphi'$ does not contribute to $[\![Q]\!]_F$, consequently, $\varphi$ is Result-Aware.

$\square$

