# OpenReview forum: "FedUP: Querying Large-Scale Federations of SPARQL Endpoints"
_ACM.org/TheWebConf/2024/Conference — TheWebConf24 Oral_

### Official Review · Reviewer_rJQV · 2023-11-17

**Novelty:** 4
**Technical Quality:** 6

**Review:**

Review:

# Overview:
The authors tackle the issue of federated query answers in SPARQL. In particular they focus on the JOIN, UNION, OPT, FILTER  fragment of SPARQL and propose to: (a) use unions over joins style of planning; and (b) filter out the parts of the union that do not contribute to query results by selecting non-empty patterns based on a sample obtained from query sources. To make the second part efficient, the proposal is to execute the pattern on the graph summary and proceed in this manner. Experimental evaluation is done over a good selection of benchmarks, with reasonable results being obtained for this sort of query planning that can be heavily skewed by the data distribution.

# Strengths:
- The idea is simple, sound, and seems to work decently well.
- There is an implementation.
- The experimental setup is decent. I particularly appreciated the fact that the authors were fair in their assesment, showing when their approach is not the best option.

# Weaknesses:
- It is a bit difficult to judge the novelty of the contribution. Namely, this just seems as another query planning paper. Everything presented seems quite reasonable to me, and I do find that the strategy of the authors has decent merits in federated query answering, but the approach can definitely suffer depending on the data distribution (particularly when the same pattern needs to be evaluated on several query sources because they do not have identical results). Perhaps a good discussion of this limitation would be better suited than the related work section which, in my opinion, is just a waste of space (as it usually is in all papers in my view).
- The writing could be improved in several aspects. While the base algorithm is quite clear, several concepts are not explained very well. In particular, the part on query summaries relies exclusively on previous work and should be expanded a bit. Next, terms like RSA, CostFed, etc. just pop out of nowhere, and might leave non experts wondering what this is about. The main example (of how Algorithm 1 works) could also use some work.
- There is a comment on parallelizing the workload, but it is never discussed on how this interacts with the proposal (it should be great since an union is computed, but this should be explained somewhere).
- A small fragment of SPARQL is covered, but this is OK for starting this line of work.
- Perhaps it is worth commenting that for some queries unraveling the joins in the fashion proposed here might generate an exponential number of query components (e.g. when nesting joins with unions).

# Recommendation:
Overall I am quite positive about this paper. It proposes a simple solution that is sound and works reasonably well in some cases. It can be made run poorly by designing the data/queries in a particular fashion, but this is true for every query planning approach. The experiments are also quite fair about this and do not show the proposed approach as winning all the time.

#Post Rebuttal update:
I would like to thank the reviewers for their reply, which answered my questions in sufficient detail. Overall, my recommendation is to accept the paper. If the paper is accepted, I would suggest briefly discussing the potential drawbacks of the approach based on differing data distributions, or the unavailability of summaries on the endpoint side, but this can probably be addressed with a few extra lines of text.

# Some typos:
- The last paragraph on page 6 is repeated.
- tp4 and tp5 in the introduction text are mislabeled.

**Questions:**

- The approach assumes that graph summaries can be computed. Would this assume that all endpoints have these published with their dataset?
- Would it be possible to have a "real-world" example in the query where federation is done with some famous endpoints such as Wikidata or DBPedia?

**Ethics Review Description:**

Nothing wrong as far as I can tell

**Reviewer Confidence:**

3: The reviewer is confident but not certain that the evaluation is correct

**Scope:**

4: The work is relevant to the Web and to the track, and is of broad interest to the community

---

### Official Review · Reviewer_Yjdz · 2023-11-22

**Novelty:** 5
**Technical Quality:** 7

**Review:**

The paper touches upon the problem of federating SPARQL queries over multiple endpoints without the user explicitly delcaring the federation. It proposes a method for assessing the likelihood of endpoints returning a non-empty solution, which seems to be an evolution of the one originally in FedX, the latter also serving as an implementation basis. The novelty seems to lie in the way this affects query planning.

First it should be said that this work lies between the field of database systems and that of semantic technologies: as such, on might fail to see its intrinsic relation to the Web per se. However, I believe this paper's position is defendable, first because SPARQL federation assumes the Web as a natural medium, and second, because RDF datasets are intended to share common structures for their global understanding across the Web - as well-illustrated in the part of the evaluation that uses FedShop.

Having said that, the authors are intellectually honest in their evaluation, about proposing an improvement that assumes a certain scale to be in place, and that is not expected to always outperform the state of the art on smaller scales. Federating over many endpoints is a huge practical limitation of Linked Open Data, and this work is well-placed in offering a sound approach, whose code they have also cared to submit to Anonymous GitHub (they even anonymised Java package names which is appreciated).

The paper also reads very smoothly, is complete with examples including leading ones, and uses appendices appropriately (for proofs but not for artificially extending paper length, well done).

A few minor items to address on an otherwise well-written paper:
- L338: "existing source selection do not" -> "does not" (or "elections do not"?)
- L803: "by order of magnitutes" -> "by orders of magnitude" (or "by an order of magnitude"?)
- L818-19: "others are zero-knowledge, *whereas* FedX or Lusail..." (if I have well understood)
- L933 (appendix): "It does not holds that" -> "hold"
- Appendices: where you use "it exists [variable]" it is generally better to say "there exists" instead

**Questions:**

- More a curiosity: this approach takes into account endpoints that are more likely to return a result. Is it also / should it also be aware of which ones would expand the result set more than others, and how?

**Ethics Review Description:**

not flagged

**Reviewer Confidence:**

3: The reviewer is confident but not certain that the evaluation is correct

**Scope:**

3: The work is somewhat relevant to the Web and to the track, and is of narrow interest to a sub-community

---

### Official Review · Reviewer_seoz · 2023-11-22

**Novelty:** 4
**Technical Quality:** 6

**Review:**

This paper presents an approach for federated processing of SPARQL queries based on the concept of result-aware query plans.
The principle behind this approach is to ensure that all subqueries contribute results to the aggregated one, thus avoiding wasted resources on endpoints that anyways do not provide matching triples. The mechanism uses provenance tracking of results in order to build the federated plans. Experimental evaluation shows promising results compared to existing alternatives in two benchmarks.

The main contributions are related to the optimisations performed on the query plan regarding the join over union problem, and most specifically the computation of query plans based on quotient summaries. However, as the authors acknowledge the main benefits of this approach are for the moment focused on the scaling of the federation. Nevertheless, the provided experiments are limited to 20-200 scenarios, but there is no clear indication of why these are the chose numbers. What happens if SPARQL federations go to 1K endpoints? Is the scaleup linear for the given approach? Is it even worth looking at that level of scale, or is 200 large enough? More informed decisions on the experimental setup would be valuable, or more evidence showing how the complexity may grow.
Given that most of the interesting improvements are shown in FedShop, it also remains unclear if there can also be a benchmark-oriented bias. Could it be the case that in other type of settings the results would be mitigated? Many of the SPARQL federation issues come from problems of real world environments, where network latency, differences among endpoint capabilities, errors, etc. play a major role. One may argue that those aspects are out of scope, but they also fall into the source selection problem, so there should be room to at least discuss the interrelationships.
This also links with potential problems depending on the dynamicity of the data source. Is the approach robust to significant changes in the query workload? Is evolution of the data patterns potentially a problem for maintaining the query plans?
Although the approach is well formalised, the cost of maintaining these plans and the provenance is not too well described. Is the computation of the quotation summaries to be taken into account for the cost of query processing? Perhaps not a critical point, but all costs need to be analysed.
Finally, the evaluation conditions seem to assume certain homogeneity in the endpoints, which might differ form more realistic conditions.
Minor: RSA, the best resource assignment should be described in the text, and explained for people not familiar with FedShop

**Strengths**
 - Implementation provided and code/repo given
 - evaluation using existing benchmarks
 - fairly complete formalisation of the approach

**Weaknesses**
 - somehow narrow scope regarding endpoint scalability
 - evaluation limitations could be clarified

**Clarity**
Fairly well written paper, easy to follow.

**Questions:**

- What are the motivations for the experimental setup, what are reasonable scaling numbers for a federation
- Is there risk of bias regarding the benchmark? Would there be interest in exploring more heterogeneous environments for evaluation?
- How does the approach behave if query workloads have significative changes? Are result-aware query plans easily maintainable? Are there challenges related to evolving data sources?
- Is the overhead of provenance management negligible? What are the costs, and is it worth evaluating them?

**Reviewer Confidence:**

3: The reviewer is confident but not certain that the evaluation is correct

**Scope:**

4: The work is relevant to the Web and to the track, and is of broad interest to the community

---

### Official Review · Reviewer_S3Cd · 2023-11-24

**Novelty:** 3
**Technical Quality:** 3

**Review:**

This paper proposes FedUp, a federation engine over multiple SPARQL endpoints. The main motivation of the paper is that existing SPARQL endpoint federation engines perform triple-pattern-wise source selection, which is not  final result-aware. As such, many sources (SPARQL endpoints) can be relevant to the  individual triple patterns but they might not contribute to the final result set. Consequently, it is highly possible that a large number of intermediate results are retrieved which are excluded after performing all the required joins to compute the final result set of the query. The author solved this problem by using index-aware source selection combined with SPARQL ASK queries. The author also claims that existing source selection algorithms do not produce union-over-join plans. They used quotient summaries as index. The proposed approach is compared with state-of-the-art federation engines using both real-world (LargeRDFBench) and synthetic (FedShop) benchmarks. The evaluation result shows the superiority of the proposed approach when it comes to federated SPARQL query processing over large endpoints.

Overall, the paper topic is very interesting and perfectly fits into the conference topics. The paper is easy to follow. The results are promising. However, there are several claims that need proper justifications and more explanations. Some of my concerns are given below and also summarized in the questions section, which I am expecting to be considered in the rebuttals.

W1. The main motivation of the paper highlighted in the abstract, section 1 and section is follows.

Abstract.  “One major issue comes from the current definition of the source selection problem, i.e., finding the minimal set of SPARQL endpoints to contact per triple pattern. Even if such a source selection is minimal, only a few combinations of sources may return results. Consequently, most of the query processing time is wasted evaluating combinations that return no results. In this paper, we introduce the concept of Result-Aware query plans. This concept ensures that every subquery of the query plan effectively contributes to the result of the query”

Section1. “One major issue comes from the current definition of the source selection problem, i.e., finding the minimal set of SPARQL endpoints to contact per triple pattern [19].”

Section 2. “In summary, the current source selection definition hides important information about which sources should be combined to find results. With just a set of relevant sources per triple pattern, it is impossible to know which combinations of sources contribute to the final results of queries”
“Example 2 (Joins-over-unions logical plans).
All existing federation engines generate such joins-over-unions plans [7].”

“As stated in the previous section, existing source selection does not reveal which combinations of relevant sources effectively produce results. Without this information, a class of query plans cannot be explored, such as the union-over-join query plans”

⇒ I am afraid, this is not the right claim as the major aim of the source selection described in [19] was to propose join-aware source selection (which is termed as result-aware source selection this paper), i.e., select those sources which are not only relevant to the individual triple patterns but also to the final result set of the query.

The problem statement in [19] is defined as follows

“Definition 2 (Optimal source Set). The optimal source set Oi ⊆ Ri for a triple pattern tpi ∈ T P contains the relevant sources d ∈ Ri that actually contribute to computing the complete result set of the query.
Definition 3 (Problem Statement). Given a set D of sources and a query q, find the optimal set of sources Oi ⊆ D for each triple pattern tpi of q. “

Clearly, the goal was not only triple-pattern-wise source selection. Rather, the goal was to perform result-aware source selection. As such, the main motivation or the problem statement is already tackled in [19] and hence it is important to make a clear distinction between the main problem statement of the proposed work w.r.t [19].


W2.  The approach lacks explaining core steps of the algorithms. For example,


𝑠𝑜𝑙𝑠(𝜑) as a function that returns the mappings resulting in the evaluation of the expression 𝜑 over  F.
How is this step done? The mappings are obtained using only the summary or actually contacting the endpoints?  It is mentioned later in the evaluation that SPARQL ASK queries are also used. However, it is not explained in detail when exactly the SPARQL ASK queries are used and when only the summaries are sufficient? Is it also checking if a join will produce any results? If this is the case, I was wondering how the proposed approach is different from LUSAIL [1], which also performs ASK queries to check which joins over a subset of sources are actually producing results.

W3. The intuition behind 𝜓ℎ is that authorities alone allow federation engines to identify which endpoints host a specific triple.

I am not sure if this can be only done by only using quotient summaries. For example

Endpoint data.
                                       <http://dbpedia.org/resource/A>   :p1     :o1 .




The authority is  dbpedia.org. However, only using authorities we cannot determine if the following  triple-pattern will yield some result or not.

            Select ?o1 Where
                  {
                     <http://dbpedia.org/resource/A>  ?p1  ?o1

                  }

W4. The paper seems to be more focused on the source selection. I missed how the query plans are generated? How the joins ordering is performed ? How the joins are actually executed, i.e. which physical joins are used (e.g. nested loop, hash, bind etc.).

W5. The author also claims that  existing source selections like HiBISCUS and Lusail do not produce union-over-joins plans. However, It is not discussed how and why they are not producing such plans? BTW, the aim of both the source selection or query planning is to produce result-aware query plans and not just triple pattern-wise source selection.

W6.  The proposed approach uses quotient summaries from [4], combined with SPARQL ASK queries. I was wondering how the proposed summaries are better than costfed summaries ? The use of SPARQL ASK queries combined with index for source selection is already used in CostFed and HiBiSCUS. Lusail uses SPARQL ASK queries for a set of triple patterns to actually check if a subset of triple patterns can produce any results. As such, the novelty of this approach needs to be clearly stated.

W7. In my opinion, the main contribution of this paper is the result-aware source selection. Unfortunately, the efficiency of the proposed source selection is not sufficiently compared with state of the art. For example, the LargeRDF bench used various metrics to exclusively compare the efficiency of source selection. I believe the results pertaining to some of these metrics would have strengthened the claims made by the author.  For example,  for each query, it would have been very interesting to report the total number of sources selected (both distinct or triple-pattern-wise) by the proposed approach compared to state of the art. From this result, we could clearly see how efficient the proposed source selection is and how much irrelevant sources are excluded. In addition, the source selection time is a very relevant measure. Finally, the number of SPARQL ASK requests used during the source selection could be interesting as well.

**Questions:**

Q1.  While both Hibiscus and Lusail do not perform only triple-pattern-wise source selection, I was wondering how the claims highlighted in W1 are true?

Q2. How 𝑠𝑜𝑙𝑠(𝜑) function is actually solved? I.e. when quotient summaries are used and when SPARQL ASK queries are required?

Q3.  The intuition behind 𝜓ℎ is that authorities alone allow federation engines to identify which endpoints host a specific triple. How can this be ensured for triple patterns with variable predicates and bound subjects?

Q4. The paper was more focused on the source selection. How is the final query execution performed? How join ordering and physical join execution is performed?

Q5. I failed to understand how LUSAIL, HIBISCUS do not produce union-over-joins plans. It would be very interesting to discuss this in detail, as this is considered as one of the main motivations of this paper.

Q6. The proposed approach uses summaries from [4] and also made use of the SPARQL ASK queries. As such, the novel contribution of this paper needs to be emphasized.

Q7. Results of metrics pertaining to the efficiency of the source selection are missing.

**Ethics Review Description:**

Nothing

**Reviewer Confidence:**

4: The reviewer is certain that the evaluation is correct and very familiar with the relevant literature

**Scope:**

4: The work is relevant to the Web and to the track, and is of broad interest to the community

---

### Official Review · Reviewer_u9uM · 2023-12-13

**Novelty:** 6
**Technical Quality:** 7

**Review:**

The paper proposes an approach for efficient querying over a large federation of SPARQL endpoints.
The authors argue that the strategy used in state-of-the-art solutions to solve the critical problem of minimal source selection obfuscates the the query optimizers ability to identify the best query plans and produces plans in which a lot of subplans are empty.
The authors introduce the concept of Result-Aware query plans that ensures that every subquery of a generated query plan is productive, leading to time savings from avoiding wasteful subquery evaluations. More specifically, it was argued that existing federated query plan generators use join-over-union style query plans which was shown to be less effective than union-over-join style plans. Some logical query equivalence rules were proposed and a concept of a result-awareness property.  These concepts formed the foundation for a problem result-aware source selection algorithm which builds union-over-join plans that have only expressions that contribute to the final results of a query. In order to break the cyclic dependence between source selection and getting query results, the authors introduce a concept of quotient summaries of federations over which the source selection algorithm can be run. This introduction of quotient summaries trades off accuracy of source selection for performance.
The overall approach proposed in encapsulated in a federation engine FedUP that produces Result-Aware query plans by tracking the provenance of query results.
Experimental evaluation involved comparing against two top leading federation engines using two benchmarks with very promising results.

+ problem well motivated and relevant to the community
+ paper is well executed with good illustrative examples.
+ approach and rationale are well presented
+ experimental plan seems adequate and results appear to be very promising

- minor error on page 1; I believe should be tp5 and tp6

**Questions:**

None

**Reviewer Confidence:**

3: The reviewer is confident but not certain that the evaluation is correct

**Scope:**

4: The work is relevant to the Web and to the track, and is of broad interest to the community

---

### Decision · Program_Chairs · 2024-01-22

**Decision:**

Accept (Oral)

**Comment:**

This article introduces FedUp, a new SPARQL federation engine that optimizes source selection using result-aware query plans.
 Results show that this approach outperforms the state of the art when querying over a large number of endpoints.

 All reviewers agree that this work is interesting and relevant to the Web Conference, and deserves to be accepted.
 We recommend the authors to incorporate the comments and clarifications that arose during the discussions, especially those by reviewer S3Cd.